# Structure of the catalytically active APOBEC3G bound to a DNA oligonucleotide inhibitor reveals tetrahedral geometry of the transition state

Atanu Maiti [1,5], Adam K. Hedger [2,3,4,5], Wazo Myint [1], Vanivilasini Balachandran[1], Jonathan K. Watts [3,4], Celia A. Schiffer [2,4] & Hiroshi Matsuo [1] ✉

APOBEC3 proteins (A3s) are enzymes that catalyze the deamination of cytidine to uridine in single-stranded DNA (ssDNA) substrates, thus playing a key role in innate antiviral immunity. However, the APOBEC3 family has also been linked to many mutational signatures in cancer cells, which has led to an intense interest to develop inhibitors of A3's catalytic activity as therapeutics as well as tools to study A3's biochemistry, structure, and cellular function. Recent studies have shown that ssDNA containing 2′-deoxy-zebularine (dZ-ssDNA) is an inhibitor of A3s such as A3A, A3B, and A3G, although the atomic determinants of this activity have remained unknown. To fill this knowledge gap, we determined a 1.5 Å resolution structure of a dZ-ssDNA inhibitor bound to active A3G. The crystal structure revealed that the activated dZ-$H_2O$ mimics the transition state by coordinating the active site $Zn^{2+}$ and engaging in additional stabilizing interactions, such as the one with the catalytic residue E259. Therefore, this structure allowed us to capture a snapshot of the A3's transition state and suggests that developing transition-state mimicking inhibitors may provide a new opportunity to design more targeted molecules for A3s in the future.

The apolipoprotein B mRNA editing enzyme catalytic polypeptide-like (APOBEC) family of proteins are cytosine deaminases which convert cytosine to uracil in DNA and RNA[1,2]. APOBEC1 is the founding member of this family, described to deaminate a cytidine in the apolipoprotein B pre-mRNA in plasma metabolism almost three decades ago[3–5]. Within this larger family, APOBEC3 proteins (A3s) catalyze the deamination of cytidines in single-stranded DNA (ssDNA) providing anti-viral activities in the innate immune response[6–11]. However, while those A3s play a beneficial role in anti-viral immunity, they may lead to drug resistance[12–15]. Furthermore, A3A and A3B have been found to be a source of cancer-associated mutations in various types of cancer such as breast, bladder, head and neck, cervical, and lung cancer. Recent studies have shown that the dysregulation of A3 proteins is a major endogenous source of DNA mutations in approximately 75% of cancer types and >50% of all cancers analyzed[16,17]. A3 proteins can cause DNA mutations either alone or as a response to cancer therapies which can drive evolution of cancers, and A3 related mutations have been associated with poor prognosis and therapeutic resistance in cancers[18–27].

[1]Cancer Innovation Laboratory, Frederick National Laboratory for Cancer Research, Frederick, MD, USA. [2]Institute for Drug Resistance, University of Massachusetts Chan Medical School, Worcester, MA, USA. [3]RNA Therapeutics Institute, University of Massachusetts Chan Medical School, Worcester, MA, USA. [4]Department of Biochemistry and Molecular Biotechnology, University of Massachusetts Chan Medical School, Worcester, MA, USA. [5]These authors contributed equally: Atanu Maiti, Adam K. Hedger. ✉e-mail: hiroshi.matsuo@nih.gov

This propensity for driving the evolution of cancers and the involvement in antiviral drug resistance makes specific inhibition of these enzymes highly desirable.

Although interest in targeting APOBEC3 has intensified, screening efforts to find small molecule inhibitors have only yielded weak binding compounds that inhibit A3's at low micromolar concentrations[28–31]. More promising results emerged from efforts to develop sequence selective inhibitors of A3 enzymes using 2′-deoxy zebularine (dZ) incorporated into short oligonucleotides[32,33]. The resulting ssDNA containing 2′-deoxy-zebularine (dZ-ssDNA) inhibited A3 deamination at low micromolar concentrations, and exhibited specificity for A3 proteins[32,33]. However, dZ-ssDNA oligonucleotides are not drug-like and are unstable in blood and cells, therefore further modifications are required to be useful as tools to study biology of A3s in cells, or lead compounds for drug discovery and development campaigns. Therefore, further efforts are needed to better understand key determinants of dZ-ssDNA binding to A3s, and use that information to develop the next generation of potent A3 specific inhibitors.

In general, structural analysis is a powerful strategy to visualize how proteins engage their ligands and binding partners, thus providing a framework for understanding specificity and affinity of binding, as well as informing drug and tool compound design. For example, co-crystal and NMR structures of A3A[34–36] and chimeric A3B-CTD (catalytic domain)[35] with a ssDNA substrate bound have provided information with regard to the 5′-TC target sequence preference for A3A and A3B. The structure of A3G-CTD2 co-crystalized with substrate ssDNA provided information on how A3G-CTD specifically recognizes its 5′-TCCCA preferred target sequence[37]. However, to date, there are no structures of an A3 enzyme complexed with a competitive active site inhibitor. Here, we present the 1.5 Å resolution co-crystal structure of a soluble variant of human A3G catalytic domain (A3G-CTD2) bound to a 9-nucleotide ssDNA inhibitor containing 2′-deoxy-zebularine at the target position, 5′-AATCCdZAAA (dZ-ssDNA). This high-resolution structure clearly showed that dZ was hydrated, generating a tight-binding tetrahedral intermediate that coordinates with the active site $Zn^{2+}$ ion. Thus, our structure captured the transition state, suggesting that hydrated dZ (dZ-$H_2O$) mimics the transition state of the A3's deamination mechanism. Thus, our results lay the foundation for designing further inhibitors for A3 enzymes.

## Results

### Synthesis of dZ phosphoramidite and dZ-containing oligonucleotide

We synthesized the 2′-deoxy-zebularine (dZ) phosphoramidite according to previous literature[32]. Briefly, the 2-hydroxypyrimidine nucleobase was silylated and then glycosylated to 2′-deoxy-3′,5′-di-O-p-toluoyl-α-D-erythro-pentofuranosyl chloride (Hoffer's chlorosugar). The 3′- and 5′- protecting group esters were removed using ammonia, to give an anomeric mixture of 2′-deoxy-zebularine nucleosides. After protecting the 5′-OH group with a dimethoxytrityl (DMT) group, we isolated the desired β anomer using column chromatography. This product was then 3′-O-phosphitylated using 2-cyanoethyl-N,N-diisopropylchlorophosphoramidite (see methods for further details and characterization). We then incorporated this dZ monomer into a short DNA oligonucleotide (5′-AATCCdZAAA) using an AKTA Oligopilot10 at a 12 μmole synthesis scale in place of the target cytidine, in the preferred 5′-CCCA- binding motif for A3G. The oligonucleotide sequence used in this work matches that of our previous work[37,38].

### Activated 2′-deoxy-zebularine generates strong affinity to catalytically active A3G-CTD

We tested the binding affinity of the 9 nt dZ-oligonucleotide (DNA: 5′-AATCCdZAAA) for the catalytically active and soluble A3G-CTD variant (referred to throughout as A3G-CTD2)[37], or the catalytically inactive A3G-CTD2 mutant (E259A mutation, referred to throughout as A3G-CTD2*). For the active A3G-CTD2, the apparent dissociation constant, $K_D$, was determined to be $0.27 \pm 0.07$ μM (Table 1 and Supplementary Fig. 2a), whereas the $K_D$ for the catalytically inactive A3G-CTD2 mutant was determined to be $14.3 \pm 1.6$ μM, representing a ~50 fold drop in affinity. Furthermore, the $K_D$ for the equivalent DNA substrate containing 2′-deoxy-cytidine instead of dZ, (DNA: 5′-AATCCCAAA), to the inactive A3G-CTD2* was determined to be $5.8 \pm 0.8$ μM.

We also tested the ability of the 5′-AATCCdZAAA oligonucleotide to inhibit deamination of the equivalent substrate by A3G-CTD2, using a [1]H-based NMR assay[37,39]. We determined the apparent $K_m$ and $V_{max}$ for the substrate at 200 nM enzyme concentration to be $509 \pm 86$ μM and $33.4 \pm 4.4$ min$^{-1}$ respectively (Table 2). In the presence of 50 μM 5′-AATCCdZAAA oligonucleotide inhibitor, the apparent $K_m$ increased to $8.8 \pm 1.8$ mM and the $K_i$ was measured to be $3.07 \pm 0.78$ μM (Table 2). While the $K_D$ value ($0.27 \pm 0.07$ μM) and $K_i$ value ($3.07 \pm 0.78$ μM) are not exactly the same, this is likely due to variations in experimental conditions which include varied protein concentration and equilibration time. We confirmed the inhibition mode to be competitive through a Lineweaver–Burk plot (Supplementary Fig. 2b). These data indicate that the 5′-AATCCdZAAA oligonucleotide is a potent inhibitor for the deamination reaction of A3G.

### Co-crystal structure of A3G-CTD2 and 5′-AATCCdZAAA

We solved the high-resolution co-crystal structure of A3G-CTD2 bound to our dZ-oligonucleotide inhibitor characterized above (5′-AATCCdZAAA). The structure was determined to a resolution of 1.5 Å by molecular replacement using a previously determined structure of the catalytically inactive A3G-CTD2*:ssDNA substrate complex (PDB 6BUX). The final refinement of the structure resulted in $R_{work}/R_{free}$ of 0.17/0.19, respectively (Table 3). The structure was solved in the space group P2₁ and contains a single A3G-CTD2:dZ-ssDNA complex in the asymmetric unit, similar to the previous co-crystal structure of A3G-CTD2* with ssDNA substrate (Fig. 1a). Previously for the inactive A3G-CTD2*:ssDNA structure, we used 2′-deoxy-cytidine at the deamination target site ($C_0$) and replaced the catalytic residue E259 with alanine to prevent catalysis[37]. In the new structure, we used active protein retaining E259, and replaced the target cytidine by a cytidine analog, 2′-deoxy-zebularine. This structure of an oligonucleotide inhibitor bound to fully catalytically competent (active) A3G, and, at 1.5 Å, represents the highest resolution A3 complex solved to date.

Given that our structure features the active A3G enzyme, we observed that 2′-deoxy-zebularine was converted to a tetrahedral hydration product; 4-(R)-hydroxy-3,4-dihydro-2′-deoxy-zebularine (dZ-$H_2O$). In this hydration product, the N3 nitrogen of the zebularine nucleobase has become protonated, and a hydroxyl group added to the top face of the C4 carbon (Fig. 1b). Direct coordination of the C4-OH to the $Zn^{2+}$ ion is evident by the short $Zn^{2+}$-O distance (2.01 Å) and

## Table 1 | $K_D$ values from Micro Scale Thermophoresis

| Protein | Ligand | $K_D$ |
|---------|--------|-------|
| CTD2 | 5′-AAT CCdZ AAA | $0.27 \pm 0.07$ μM |
| CTD2* | 5′-AAT CCdZ AAA | $14.3 \pm 1.6$ μM |
| CTD2* | 5′-AAT CCC AAA | $5.8 \pm 0.8$ μM |

## Table 2 | Enzymatic parameters for A3G CTD2

| Substrate | Inhibitor | $K_m$ | $V_{max}$ | $K_i$ |
|-----------|-----------|-------|-----------|-------|
| 5′-AAT CCC AAA | n/a | $0.509 \pm 0.086$ mM | $33.4 \pm 4.4$ min$^{-1}$ | n/a |
| 5′-AAT CCC AAA | 5′-AAT CCdZ AAA | $8.8 \pm 1.8$ mM | $42.3 \pm 8.6$ min$^{-1}$ | $3.07 \pm 0.78$ μM |

**Table 3 | Data collection and refinement statistics**

| Data collection | |
| --- | --- |
| Space group | P2₁ |
| Cell dimensions | |
| a, b, c (Å) | 47.48, 47.12, 52.04 |
| α, β, γ (°) | 90.00, 103.18, 90.00 |
| Resolution (Å) | 50.00–1.50 (1.55–1.50)* |
| $R_{merge}$ (%) | 8.6 (46.8) |
| $R_{meas}$ (%) | 10.4 (56.1) |
| $R_{pim}$ (%) | 5.8 (30.6) |
| I/σI | 12.91 (1.72) |
| CC1/2 | 0.963 (0.752) |
| Completeness (%) | 93.4 (98.9) |
| Redundancy | 3.2 (3.2) |
| *Refinement* | |
| Resolution (Å) | 34.50–1.50 (1.55–1.50) |
| No. of reflections | 33634 (3534) |
| $R_{work}$/$R_{free}$ (%) | 17.25/19.44 |
| No. of atoms | 1933 |
| Protein | 1539 |
| DNA | 179 |
| Ligand/ion | 1 (Zn²⁺) |
| Water | 212 |
| *B-factor* | |
| Average B-factors (Å²) | 36.70 |
| Protein/DNA | 35.70 |
| Ligand/ion | 27.60 |
| Water | 44.40 |
| *r.m.s deviations* | |
| Bond lengths (Å) | 0.005 |
| Bond angles (°) | 0.761 |

*Values in parentheses are for highest-resolution shell

continuous electron density around the Zn coordination sites (Fig. 1b). Furthermore, we observed clear density of the key catalytic E259, with the sidechain forming hydrogen bonds with dZ-H₂O (Fig. 1c). These results are in general agreement with the deamination mechanism proposed for bacterial cytidine deaminases (CDAs), related enzymes that deaminate single cytidine/2′-deoxycytidine[40–50]. Analogous to the CDAs, A3s also depend on a Zn²⁺ located at the center of the substrate binding pocket (Fig. 1a), and the deamination reaction proceeds via a tetrahedral intermediate (Figs. 1c and 2). Therefore, our use of the active A3G-CTD2 for structural analysis has enabled the capture of the transition state, thus providing support for the proposed tetrahedral intermediate and offering an unprecedented view of the A3 catalytic mechanism[49,50].

## Comparison between active A3G-CTD2:dZ-ssDNA and inactive A3G-CTD2*:ssDNA structures

Backbone superposition of A3G-CTD2 (catalytically active, this structure) and A3G-CTD2* (catalytically inactive, 6BUX) bound to ssDNA showed that the structures are almost identical, with a root mean square deviation (RMSD) for CA backbone atoms of 0.2 Å (superimposition is shown as Supplementary Fig. 3). Like our previous structure, all nine nucleotides of ssDNA including 2′-deoxy-zebularine (dZ) are well ordered and all five substrate specific nucleotides (5′-$T_{−3}C_{−2}C_{−1}dZ_0A_{+1}$) form direct interactions with A3G. These interactions between the nucleotides flanking the active site ($T_{−3}$, $C_{−2}$, $C_{−1}$ and $A_{+1}$) are essentially conserved (Fig. 3a, b). However, within the active site we see significant differences, including around the dZ nucleobase and Zn²⁺ coordination residues (Fig. 3c).

In the A3G-CTD2:dZ-ssDNA structure of the catalytically active complex, the essential catalytic residue E259 is oriented towards the zebularine nucleobase as expected (Fig. 1c), poised to carry out its acid/base role during catalysis. The side chain carboxylate of E259 makes two hydrogen bonds, to both the N3-H and Zn-coordinated C4-O (Fig. 3a). The oxygen of the C4-OH, is directly coordinated to the 4th Zn-coordination site, replacing the Zn-bound water molecule seen in the previous structure of inactive A3G. This provides direct structural evidence of the second step of the deamination mechanism (Fig. 2f), whereby the Zn-coordinated hydroxide has attacked the top face of the 2′-deoxy-zebularine C4 carbon, forming a tetrahedral transition state intermediate (Fig. 2g). To make this direct C4-O-Zn coordination

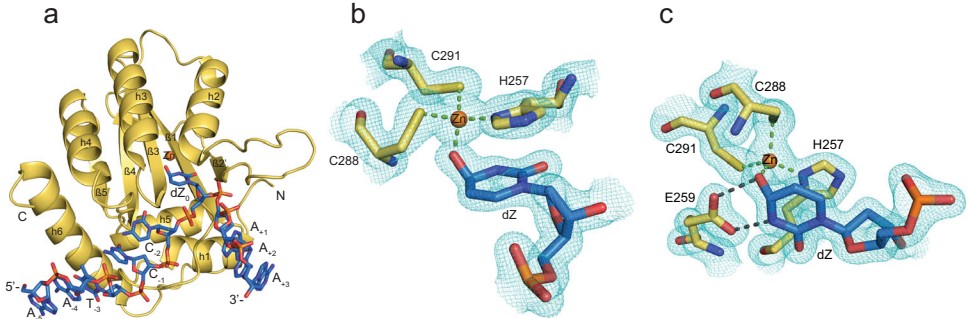

**Fig. 1 | Structure of active APOBEC3G catalytic domain (A3G-CTD2) with ssDNA containing 2′-deoxy-zebularine at the target site. a** The asymmetric unit contains one protein shown as cartoon (yellow) and one oligonucleotide inhibitor (blue) shown as sticks. N and C indicate the N- and C-terminal ends of the protein, 5′- and 3′- indicates 5′ and 3′ ends of the oligonucleotide. **b** A 2Fo–Fc electron density map contoured at 1.5 σ is shown in cyan around the 2′-deoxy-zebularine hydrated intermediate (dZ-H₂O), the Zn²⁺ ion, and the Zn²⁺ coordinating residues. The 3′-oxygen of the preceding nucleotide of dZ- is not shown. Zn²⁺ is shown as an orange sphere. C, N, S and O atoms are colored yellow, navy blue, gold, and red respectively, for amino acid residues of the protein. Atoms in nucleotides are colored blue, navy blue, red, and orange for C, N, O, and P, respectively. Green dotted lines indicate the coordination bond with the Zn²⁺ ion. Continuous electron density around the 2′-deoxy-zebularine hydrated intermediate, Zn²⁺ ion and Zn²⁺ coordinating residues support the formation of the 4-(R)-hydroxy-3,4-dihydro-2′ deoxy-zebularine intermediate where oxygen of C4-hydroxy directly coordinates with Zn²⁺ ion. **c** The structure was rotated about 90 degrees from (**b**) along the axis of the 2′-deoxy-zebularine ring plane. A 2Fo–Fc electron density map showing E259 with the 2′-deoxy-zebularine hydrated intermediate, Zn²⁺ and Zn²⁺ coordinating residues. Gray dotted lines indicate hydrogen bonds between E259 and dZ-H₂O.

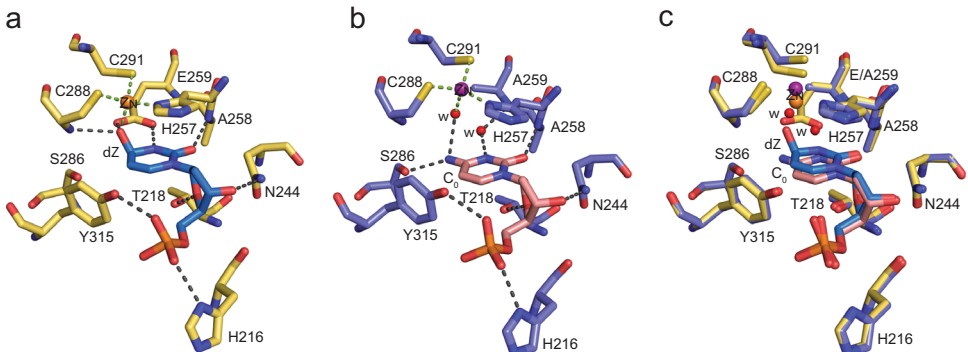

**Fig. 2 | Mechanism of deamination by A3 enzymes on dC and dZ showing key tetrahedral intermediates.** Top panel, cytidine nucleobase; **a** Key active site Glu protonates the N3 nitrogen and absorbs a proton from the Zn-coordinated water molecule. **b** Zn-coordinated hydroxide ion attacks the activated C4 carbon as the Glu protonates the C4-amine. **c** The key tetrahedral intermediate collapses as the Glu absorbs the C4-OH proton, a carbonyl is formed, and ammonia leaves. **d** Products formed (uracil and ammonia) after deamination is complete. Bottom panel, 2′-deoxy-zebularine nucleobase; **e** Same as for cytidine (panel **a**). **f** Same as for cytidine (panel **b**) but there is no C4-amine to be protonated. **g** Due to no C4-amine and the poor leaving group ability of H, the final product is hydrated-dZ (dZ-H₂O, having added OH at C4 and H at N3) - a transition state mimic, which remains bound to the enzyme.

**Fig. 3 | Visualization of the target nucleotide in the active site and comparison with substrate (2′-deoxy-cytidine) bound to inactive enzyme.** The green and gray dotted lines indicate Zn²⁺ coordination and hydrogen bonds, respectively. **a** Active site interactions of active A3G-CTD2 with the 2′-deoxy-zebularine hydrated intermediate (dZ-H₂O) (PDB 7UXD). Zn²⁺ is shown as an orange sphere. C, N, S and O atoms are colored yellow, navy blue, golden-yellow, and red, respectively, for amino acid residues of the protein. Atoms in nucleotides are colored blue, navy blue, red, and orange for C, N, O, and P, respectively. **b** Active site interactions of inactive A3G-CTD2* with 2′-deoxy-cytidine at target position (PDB 6BUX). The Zn²⁺ and waters are shown as purple and red spheres, respectively. C, N, S and O atoms are colored light blue, navy blue, golden-yellow, and red, respectively, for amino acid residues of the protein. Atoms in nucleotides are colored pink, navy blue, red, and orange for C, N, O, and P, respectively. **c** Superimposition of two structures. Most of the active site interactions are conserved. However, significant differences in the plane of the target nucleotide's aromatic ring and Zn²⁺ coordination are observed.

possible, the 2′-deoxy-zebularine nucleobase and Zn-coordinating residues adjust their positions accordingly. The pyrimidine ring of the hydrated 2′-deoxy zebularine (dZ-H₂O) tilts upward ~8° when compared to the target cytosine ($C_0$) in our previous A3G-CTD2*:ssDNA structure. This tilt increases the distance between equivalent nucleobase atoms in the two structures gradually across the base from the anomeric C1′ (0.36 Å) to the C4 atom (0.71 Å). In addition, the Zn²⁺ ion is pulled down ~0.8 Å from the inactive A3G-CTD2*:ssDNA structure. As a result, the Zn coordinating residues also shift down relative to the A3G-CTD2*:ssDNA structure, the SG atom of 288 C, SG atom of 291 C, and ND1 atom of 257H move down 0.34 Å, 0.99 Å and 0.31 Å, respectively (Fig. 3c). Overall, in the active A3G-CTD2:dZ-ssDNA structure the total Zn-O-C4 distance is 3.5 Å compared to a Zn-water-C4 distance of

5.4 Å in the inactive A3G-CTD2*:ssDNA structure. This variation makes sense as in the inactive structure the intact cytosine nucleobase is separated from the Zn²⁺ ion by the water molecule, which becomes covalently attached to dZ in the active structure. The target cytosine is further stabilized in the active structure by the side chain of the catalytic E259 forming a direct hydrogen bond with the N3 nitrogen, in the inactive complex E259 is replaced with an ordered water coordinating between N3 and the backbone NH of A259 (Fig, 3b, c). Finally, the C4-NH₂ of the target cytosine also hydrogen bonds (2.8 Å) with the backbone carbonyl of S286, which is not possible with dZ due to lack of NH₂ group. However, the C4-OH of dZ-H₂O instead hydrogen bonds (3.0 Å) with the backbone NH of C288. All other interactions within the active site of both structures are conserved.

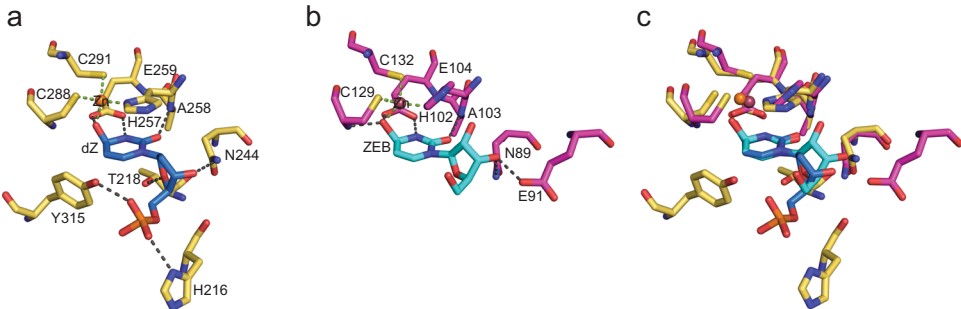

**Fig. 4 | Comparison of interactions between the catalytic site of A3G-CTD2 and *E. coli* CDA. a** Active site interactions of A3G-CTD2 with the 2′-deoxy-zebularine hydrated intermediate (dZ-H$_2$O) (PDB 7UXD). Zn$^{2+}$ is shown as an orange sphere. C, N, S and O atoms are colored yellow, navy blue, gold, and red, respectively, for amino acid residues of the protein. Atoms in nucleotides are colored blue, navy blue, red, and orange for C, N, O, and P, respectively. **b** Active site interactions of *E. coli* CDA with ribo-zebularine-hydrated intermediate (ZEB-H$_2$O) (PDB 1CTU). Zn$^{2+}$ is

shown as a raspberry sphere. C, N, S and O atoms are colored magenta, navy blue, gold and red, respectively, for amino acid residues of the protein. Atoms in nucleotides are colored cyan, navy blue and red for C, N and O respectively. **c** Superimposition of the 2′-deoxy-zebularine hydrated intermediate (dZ-H$_2$O) and the ribo-zebularine hydrated intermediate (ZEB-H$_2$O). Conserved amino acid residues are shown as stick models.

## Comparison between active A3G-CTD2:dZ-ssDNA and cytidine deaminase (CDA):ribo-zebularine structures

CDA enzymes do not bind single-stranded DNA/RNA, and have very low amino acid sequence homology when compared to A3 enzymes (Supplementary Fig. 1b). However, because both catalyze the deamination of cytosine nucleobases, key structures of active sites are conserved between CDA and A3 proteins, and their catalytic mechanisms are likely similar. This has led to the hypothesis that A3s employ the same tetrahedral transition state, as described for CDAs[42,44], and our structure now confirms this to be the case. In comparing our new A3G-CTD2:dZ-ssDNA structure with the *E. coli* CDA-ribo-zebularine structure we note the two structures show similar interactions within the active site, when aligned by overlapping the transition state mimics (Fig. 4). In both structures, the His-Cys-Cys coordination of the Zn$^{2+}$ ion is essentially identical, including direct contact with the C4-OH of the zebularine nucleobase [Zinc coordination: H257, C288 and C291 in A3G-CTD2 (Fig. 4a), and H102, C129 and C132 in *E. coli* CDA (Fig. 4b)]. The side chain carboxylic acid group of the key catalytic residue (E259 in A3G-CTD2 and E104 in *E. coli* CDA) makes two hydrogen bonds to the nucleobase; one to the N3-proton and another to the C4-hydroxyl oxygen. Similar hydrogen bonding is also observed between the zebularine-C2 carbonyl and the backbone amide NH of A258 in A3G-CTD2 and A103 in *E. coli* CDA, as well as for the C4-hydroxyl oxygen and the backbone amide NH of C288 in A3G-CTD2 and C129 in *E. coli* CDA. Finally, a hydrogen bonding interaction between the C3′-O of both ribose/2′-deoxyribose sugars and the side chain amino group of N244 in A3G-CTD2 and N89 in *E. coli* CDA is conserved. The omit map of hydrated ribo-zebularine (ZEB-H$_2$O) and hydrated-dZ (dZ-H$_2$O) contoured at 3σ shows the difference of resolution in these structures and the improved model fitting of dZ-H$_2$O (Supplementary Fig. 4). Thus our 1.5 Å resolution A3G structure provides clear evidence of hydration of dZ at the N3 and C4 positions resulting in N3-H and C4-OH, respectively, which was previously proposed as a possibility based on the 2.3 Å resolution structure of the *E. coli* CDA:ribo-zebularine.

## Discussion

APOBEC3 (A3) enzymes have been promising, but largely elusive, antiviral, and anticancer targets[7,51,52]. While small molecule inhibitor development has not been successful, modified ssDNA oligonucleotides with the C$_0$ changed to 2′-deoxy-zebularine or 2′-deoxy-zebularine analogs have been shown to inhibit A3s[32,33] although not with high potency. Zebularine was initially identified as a cytidine deaminase (CDA) inhibitor; however, CDA inhibitors do not inhibit A3s, because unlike CDAs that modify the single nucleoside, A3s

deaminate cytidine in the context of ssDNA[53], suggesting they may require additional features beyond those needed for CDA binding. The high resolution (1.5 Å) A3G-CTD2:dZ-ssDNA structure we present here now helps understand both similarities between CDAs and A3s and the observed differences. Despite the similarities of the transition state and the interactions within the catalytic site, the A3G-CTD2:dZ-ssDNA structure differs from the *E. coli* CDA:ribo-zebularine structure in several ways. *E. coli* CDA appeared to bind a single ribo-zebularine as a dimer. In fact, all CDA structures from *E. coli* to human showed dimer or tetramer association of proteins[42,44,54–56], whereas A3G-CTD2 bind dZ-ssDNA as a monomer. Furthermore, 2′-deoxy-zebularine embedded in our ssDNA inhibitor exhibits a C2′-endo DNA-like conformation, as was also observed for the C$_0$ in our previous work[37], and is consistent with the finding that the target 2′-deoxy-cytidine favors a C2′-endo sugar conformation for substrate binding and deamination[38]. Indeed, in all four *E. coli* CDA co-crystal structures with ribose-based inhibitors bound (FZEB, ZEB-H$_2$O, DHZ, and DAC), the sugar adopts a C3′-endo conformation[42,44]. It is likely that ribose-based CDA inhibitors must be converted to their C2′-endo, DNA-like, equivalents to be potent inhibitors of A3 enzymes when incorporated into ssDNA oligonucleotides. Taken together, our work provides a molecular explanation for why CDA inhibitors are inactive against A3s despite highly similar catalytic mechanisms[37,38,42,44]. The active A3G structure described here shows significant changes within the active site relative to the previously solved inactive A3G structure[37], thus providing key insights into the transition state of A3 enzymes. The catalytic glutamic acid forms two hydrogen bonds with the transition state analog at both the N3 hydrogen and the C4-hydroxyl oxygen. This hydroxyl oxygen forms the 4th binding partner of the tetrahedrally coordinated catalytic Zn$^{2+}$, replacing the coordinated water molecule seen in the inactive structure. Beyond the active site, the rest of the complex between our active and inactive A3G structures display very little conformational change as judged by the low RMSD of the backbone (0.2 Å). Furthermore, all the interactions of specific nucleotides T$_{-3}$, C$_{-2}$, C$_{-1}$ and A$_{+1}$ are essentially conserved in both structures, which suggests that we can accurately predict structural models of A3A or A3B with dZ-ssDNA inhibitors as the A3A:ssDNA and A3B:ssDNA structures are available[34–36]. These structural models with the activated dZ/H$_2$O may be used for designing inhibitors with higher affinity and specificity to A3A or A3B, as the catalytic domain of A3G recognizes nucleotides containing CC, while A3A and A3B recognizes nucleotides containing TC. Since A3A and A3B cause mutations in the human genome in cancer cells[18–27], inhibitors, with TdZ, can likely inhibit both A3A and A3B, while avoiding A3G, would be beneficial.

Our structure provides information about the inhibitor binding and the transition state of the A3s deamination mechanism. Therefore, this new structural work offers key information needed for A3 inhibitor development and helps rationalize recent activity data obtained from dZ-ssDNA based inhibitors[32,33]. For example, the substitution of the C5 hydrogen in the target dC or dZ by a methyl group (5MedC or 5MedZ) or fluorine (5FdC or 5FdZ) in ssDNA made both poorer substrates and poorer inhibitors for A3G when compared to dC or dZ respectively[32,33]. This observation was rationalized by steric effects, as Me and F are larger than H. In both of our A3G-CTD2:dZ-ssDNA and A3G-CTD2*:ssDNA structures, the positioning of the aromatic cytosine or zebularine nucleobase is perpendicular to the aromatic ring of Y315 at C5 and exhibits π−π stacking (this structure and[37]). This perpendicular arrangement likely stabilizes the substrate and transition state intermediate in the catalytic site. Introduction of Me or F at this C5 position likely destabilizes this interaction and hence precludes these from being good inhibitors. The A3G-CTD2:dZ-ssDNA structure provides clear structural evidence for the activation of dZ to a transition state analog and explains the stronger binding affinity and inhibitory activity seen for catalytically active A3 enzymes.

Transition-state analogs have been developed to target many classes of enzymes, and these compounds typically achieve very strong binding affinity and improved specificity. In fact, transition-state analogs have become powerful drugs once optimized with structure-based drug design. These include oseltamivir (Tamiflu) which inhibits neuraminidase[57–62] and treats influenza or all of the HIV viral protease inhibitors[63–72]. Despite the emerging importance of A3s as potential drug targets, they have remained recalcitrant to small molecule drug discovery. Our structure captured the transition state and revealed critical interactions stabilizing the transition state of the A3's deamination mechanism, which is a critical insight needed for designing transition-state analogs. Our high-resolution co-crystal structure of A3G-CTD2:dZ-ssDNA thus opens further opportunities to develop a new class of A3 transition state-mimicking inhibitors with therapeutic potential in oncology and infectious diseases.

# Methods

### Synthesis and Characterization of 2′-deoxy-zebularine phosphoramidite

The dZ phosphoramidite was synthesized according to previous literature[32], but was further purified by precipitation. For precipitation, the flash-purified solid (3.2 g) was dissolved in a small amount of anhydrous dichloromethane (~3 mL) and added dropwise to ~600 mL vigorously stirring cold hexane. The resulting solid was filtered and evaporated from chloroform to give the purified 2′-deoxy-zebularine phosphoroamidite, as an off-white crunchy foam. **¹H NMR** (DMSO-$d_6$, 500 MHz) was identical to that observed previously[32], **³¹P NMR** (DMSO-$d_6$, 202 MHz) δ$_{ppm}$ 147.8, 147.5 (-1:1 diastereomers), plus a small amount of P(V) by-product (H-phosphonate) at 13.9 ppm. **HRMS (ESI):** m/z Calcd. for $C_{39}H_{47}N_4O_7P$ $[M + H]^+$ 715.3255; found, 715.3250. Full NMR and HRMS spectra shown in Supplementary Figs. 5–7.

### Synthesis and Characterization of a 2′-deoxy-zebularine containing oligonucleotide

The 9 nt dZ-ssDNA oligonucleotide was synthesized at 12-µmole scale using an Akta OligoPilot synthesizer using standard methods on deoxy-adenosine (n-bz) 1000 Å 3′-lcaa controlled pore glass support (ChemGenes). DNA phosphoramidites were purchased commercially and diluted to 0.1 M in anhydrous acetonitrile (ChemGenes). The deoxy-zebularine phosphroramidite was synthesized as described above and diluted to 0.125 M in anhydrous acetonitrile. Detritylation was accomplished using 3% dichloroacetic acid in toluene (TEDIA). Oxidation was accomplished using 0.05 M iodine in pyridine/water (9:1 v/v, TEDIA). Benzylthiotetrazole was used as the activator (0.25 M in acetonitrile, TEDIA). Coupling time was 10 min. The oligonucleotide

was synthesized with 5′-DMT off, and treated with 10% diethylamine in acetonitrile while on the CPG support to remove cyanoethyl backbone protection. Deprotection was carried out using concentrated aqueous ammonium hydroxide for 8 h at room temperature. Longer exposure of dZ-containing oligonucleotides to ammonia resulted in visible side product formation by mass spectrometry. Deprotection was followed by immediate concentration using a centrifugal evaporator to remove ammonia, the aqueous solution of crude oligonucleotide was filtered (0.2 µm), and then purified by ion-exchange on a preparative scale Agilent 1260 Infinity HPLC using an Agilent PL-SAX 10 µm 1000 Å column (150 × 10 mm). HPLC was performed at 10 mL/min with a column temperature of 50 °C using a gradient of Buffer A: 70% water, 30% acetonitrile, and buffer B: 70% 1 M NaClO₄ in water containing 30% acetonitrile. UV absorbance was measured at 260 nm and the eluting oligonucleotide was fractionated and analyzed by LCMS. Fractions containing pure oligonucleotide were pooled and evaporated to dryness. The oligonucleotide was then desalted using a Nap-25 column (GE Healthcare) and evaporated to dryness again. The final dried oligonucleotide was resuspended in nuclease free water to a stock concentration of 4 mM for LCMS analysis, binding, inhibition, and crystallography experiments. LCMS characterization was carried out using a 6530 Accurate-Mass Q-TOF LC/MS (Agilent) linked to a pre-injection 1260 infinity HPLC (Agilent). Calc. MW for 5′-AATCCdZAAA oligonucleotide: 2660.8, Obs. MW: 2660.5. Full mass spectra shown in Supplementary Fig. 8.

### Protein expression and purification

All the variants of human A3G-CTD (A3G-CTD2 and A3G-CTD2*) were expressed and purified as described previously[37]. Proteins used for crystallography and NMR experiment were expressed from pGEX6P-1 expression plasmid with Glutathione S-transferase (GST) tag and proteins used for MST assay were expressed from pET-28a plasmid with poly-Histidine tag (for Ni-NTA purification) in *E. coli* BL21(DE3) cells (Invitrogen). Cells were grown in LB media at 37 °C until reaching an optical density of 0.5–0.6 at 600 nm. Then, temperature was reduced to 17 °C and protein expression was induced for 18 h with 0.2 mM isopropyl β-D-1-thiogalactopyranoside (IPTG). All the steps for protein purification were performed at 4 °C. *E. coli* cells were harvested by centrifugation and re-suspended in lysis buffer (either 50 mM sodium phosphate pH 7.3, 150 mM NaCl, 25 µM ZnCl₂, 2 mM DTT, and 0.002% Tween-20 for GST purification or 50 mM sodium phosphate pH 7.3, 150 mM NaCl, 50 µM ZnCl₂, 1 mM DTT, and 0.002% Tween-20 for Ni-NTA purification) and EDTA free protease inhibitor cocktail (Roche, Basel, Switzerland). The suspended cells were disrupted by sonication and then cell debris were separated by centrifugation at 48,384 g for 30 min. Supernatant containing desired protein was applied to either Glutathione-Sepharose resin (GE Healthcare Life Science) for GST purification or Ni-NTA Agarose resin (QIAGEN) for Ni-NTA purification, equilibrated with lysis buffer and agitated for about 2 h. For GST purification, protein-bound resin was washed with Pre-Scission Protease cleavage buffer (50 mM sodium phosphate, pH 7.5, 100 mM NaCl, 10 µM ZnCl₂, 2 mM DTT, and 0.002% Tween-20) and incubated with Pre-Scission protease (GE Healthcare Life Science) for 18 h. The supernatant containing the cleaved protein was separated from the resin by centrifugation and loaded on to HiLoad 16/600 Superdex 75 gel filtration column (GE Healthcare Life Science) equilibrated with 20 mM Bis−Tris (pH 6.5), 100 mM NaCl, 1 mM DTT, 0.01 mM ZnCl₂, and 0.002% Tween-20. For Ni-NTA purification, protein-bound resin was washed with 50 mM sodium phosphate, pH 7.3, 1 M NaCl, 25 µM ZnCl₂, 1 or 2 mM DTT, and 0.002% Tween-20. Protein was eluted from resin in buffer containing 400 mM imidazole, 50 mM sodium phosphate, pH 7.3, 100 mM NaCl, 1 mM DTT, and 0.002% Tween-20. Eluted protein was loaded on to HiLoad 16/600 Superdex 75 gel filtration column equilibrated with 20 mM Bis−Tris pH 6.5, 100 mM NaCl, 1 mM DTT, 0.002% Tween-20,

and 20 μM ZnCl₂. For both GST and Ni-NTA purification, protein purity was analyzed by SDS-PAGE.

## Microscale Thermophoresis $K_D$ measurements

The affinity of A3G-CTD2* and A3G-CTD2 for 5′-AATCCCAAA substrate and 5′-AATCCdZAAA inhibitor were determined as dissociation constants, $K_D$, using a Monolith (Nano Temper Technologies, GmbH, Munich, Germany) microscale thermophoresis instrument. RED-tris-NTA fluorescent dye solution was prepared at 100 nM in the MST buffer (20 mM Bis−Tris pH 6.5, 100 mM NaCl, 1 mM DTT, 0.002% Tween 20). A3G-CTD2* was mixed with dye at 100 nM and incubated for 30 min at room temperature followed by centrifugation at 15,000 g for 10 min. For the MST measurement, final concentrations of ssDNA solution was prepared at: 1 mM, 500 μM, 250 μM, 125.5 μM, 62.5 μM, 31.25 μM, 15.62 μM, 7.81 μM, 3.91 μM, 1.95 μM, 976 nM, 488 nM, 244 nM, and 122 nM for 5′-AATCCCAAA binding to A3G-CTD2*; 995 μM, 497.5 μM, 248.75 μM, 124.38 μM, 62.19 μM, 31.09 μM, 15.55 μM, 7.77 μM, 3.89 μM, 1.94 μM, 971.68 nM, 485.84 nM, 242.92 nM, 121.46 nM, 60.73 nM for 5′-AATCCdZAAA binding to A3G-CTD2*; 497.5 μM, 248.75 μM, 124.38 μM, 62.19 μM, 31.09 μM, 15.55 μM, 7.77 μM, 3.89 μM, 1.94 μM, 971.68 nM, 485.84 nM, 242.92 nM, 121.46 nM, 60.73 nM, 3.04 nM, 1.52 nM, 759 pM, 380 pM, 190 pM, 95 pM, 47 pM, 24 pM, 12 pM for 5′-AATCCdZAAA binding to A3G-CTD2. Final protein concentration was at 50 nM. Measurements were performed using Nano Temper MST premium coated capillaries. Measurements were performed at 23 °C with 100% excitation power and 40% MST power. The experiment was repeated three times and data analysis was carried out using MO affinity analysis software (Nano Temper Technologies).

## Real-time NMR deamination and inhibition assays

The initial rates of deamination in the absence and presence of inhibitor were performed as previously described in Maiti et al. (2018) Nature Communications 9:2460. In brief, 200 nM of A3G-CTD2 was incubated with substrate in 50 mM NaPO₄ pH 6.5, 100 mM NaCl, 0.002% Tween 20, 1 mM DTT, and 10 μM ZnCl₂. Formation of the uracil product was monitored by integration of the H5 uracil proton peak at 5.6ppm from sequentially acquired ¹H NMR spectra. Experiments were performed at 25 °C on a Bruker Avance III NMR spectrometer operating at a ¹H Larmor frequency of 600 MHz. Initial rates were measured for 5′-AATCCCAAA substrate concentrations of 100 μM, 200 μM, 400 μM, 1 mM, and 2 mM. $K_m$ and $V_{max}$ for the substrate in the absence and presence of 50 μM 5′-AATCCdZAAA were determined from the slopes and the intercepts of Lineweaver-Burk plots, and the $K_i$ was calculated from the $K_m$ values.

## Crystal growth and data collection

Active A3G-CTD2 in 20 mM Bis−Tris pH 6.5, 100 mM NaCl, 1 mM DTT, 10 μM ZnCl₂, and 0.002% Tween-20 buffer was mixed with 50% molar excess of dZ-ssDNA (5′-AATCCdZAAA) and kept at 4 °C for overnight. The mixture was concentrated to ~450 μM of protein using Amicon Ultra Centrifugal Filters unit 3KDa cutoff (Millipore Sigma). Crystals were grown at 4 °C, by sitting drop vapor diffusion method over a 300 μl reservoir of 20% W/V PEG 6000, 50 mM di-sodium L-malate; pH 5.0 and 30 mM CaCl₂ in a sitting drop 24 well crystallization plate from Hampton Research. Drops were set up by mixing 2 μl of sample and 2 μl of reservoir solution. Crystals were cryoprotected using reservoir solution containing 20% v/v glycerol and flash frozen in liquid nitrogen. X-ray diffraction data were collected at Southeast Regional Collaborative Access Team (SER-CAT) 22-ID beamline at the Advanced Photon Source, Argonne National Laboratory. The collected diffraction data were indexed, integrated, and scaled using HKL2000[73]. The crystals belong to the space group P2₁.

## Structure determination and analysis

The structure was solved at 1.5 Å resolution by molecular replacement using the program Phaser[74]. Our previous structure of A3G-CTD2* (PDB ID: 6BUX, ssDNA and solvents were removed) was used as search model. Model building of the protein and bound DNA were manually performed using the program Coot[75]. The DNA linking ligand for catalytically hydrated dZ; 4-(R)-hydroxy-3,4-dihydro-2′-deoxy-zebularine-5′-monophosphate {chemical formula: C₉ H₁₅ N₂ O₈ P and SMILES: C1[C@@H]([C@H](O[C@H]1N2C = C[C@H](NC2 = O)O)COP( = O)(O)O)O}[32] was not used before in any crystal structure. However, we found a DNA linking ligand named 3,4-DIHYDRO-2′-DEOXYURIDINE-5′-MONOPHOSPHATE (ligand ID: DDN) in PDB Ligand Library with same chemical structure (Chemical formula: C₉ H₁₅ N₂ O₈ P and SMILES: C1[C@@H]([C@H](O[C@H]1N2C = C[C@H](NC2 = O)O)COP( = O)(O)O)O). We used the ligand DDN as our DNA linking ligand for catalytically hydrated dZ (4-(R)-hydroxy-3,4-dihydro-2′-deoxy-zebularine-5′-monophosphate) in our structure. The initial and final structure refinement were performed using the programs Refmec5[76] and Phenix[77], respectively. The final model was refined to $R_{work}/R_{free}$ values of 0.17/0.19 and was validated with the PDB validation tool and Molprobity[78].

Pairwise rms deviation between A3G-CTD2 and A3G-CTD2* structure were calculated using Dali[79]. Figures of structure models were generated by PyMOL (The PyMOL Molecular Graphics System, Version 1.2r3pre, Schrödinger, LLC.) and UCSF-Chimera[80].

## Reporting summary

Further information on research design is available in the Nature Portfolio Reporting Summary linked to this article.

## Data availability

The data that support this are available from the corresponding author upon request. The atomic coordinates and structural factors for the reported crystal structure have been deposited in the Protein Data Bank under the accession number 7UXD. Source data are provided with this paper.

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

## Acknowledgements

This work was supported in part by a grant from the U.S. National Institutes of Health R01AI150478 for C.A.S. and H.M. For A.M., W.M., V.B. and H.M., this project has been funded in part with federal funds from the National Cancer Institute, National Institutes of Health, under contract 75N91019D00024. W.M. was supported in part by the NIH Office of Intramural Training and Education's Intramural AIDS Research Fellowship. A.H. was supported in part by a predoctoral fellowship from the PhRMA Foundation. The content of this publication does not necessarily reflect the views or policies of the Department of Health and Human Services, nor does mention of trade names, commercial products, or organizations imply endorsement by the U.S. Government.

## Author contributions

A.M. performed the expression and purification of protein with support from V.B.; A.M. performed the protein crystallization, X-ray data collection, crystal structure solution and refinement; A.M., A.H., C.A.S., and H.M. performed the structural analysis and interpretation; W.M. performed the MST and NMR catalytic assays and analyzed the data; A.H. synthesized the 2'-deoxy-zebularine phosphoramidite and the ssDNA oligonucleotide containing 2'-deoxy-zebularine, and J.W. supervised A.H.; A.M., W.M., A.H., C.A.S., and H.M. wrote the paper; C.A.S and H.M. conceived, coordinated and oversaw the project; C.A.S. and H.M. secured funding.

## Funding

## Competing interests

The authors declare no competing interests.
