## [Peer Review File · Nature Communications]

Structure of the catalytically active APOBEC3G bound to a DNA oligonucleotide inhibitor reveals tetrahedral geometry of the transition stateReviewers' Comments:

Reviewer #1:

Remarks to the Author:

The manuscript by Maiti et al reports a 1.5-angstrom crystal structure of the catalytic domain of human APOBEC3G (A3G-CTD) bound to a single-stranded DNA substrate containing the cytidine analog zebularine. The authors suggest the reported structure will help fill a major gap in the field of A3 enzymes: the lack of cell-permeable inhibitors that could provide useful tool compounds to test their roles in viral evolution (by sublethal mutagenesis in the case of A3G) or cancer (for A3A and A3B).

They show A3G, and likely other A3 family members, catalyze deamination through a key tetrahedral intermediate which is mimicked by deoxyzebularine incorporated into the ssDNA oligo used for cocrystallization with A3G-CTD. The authors compare their structure of A3G bound to DNA containing zebularine to a previously reported structure of *E. coli* CDA containing ribo-zebularine, showing that the transition states are essentially identical. While ribo-zebularine potently inhibits bacterial CDA on its own, flanking DNA is required for deoxy-Zebularine to inhibit A3G as explained by the authors' structure. The flanking nucleotides of Zeb-DNA adopt the same conformation as observed in prior A3G-DNA structures. The structure reported by Maiti et al is well supported by direct binding (microscale thermophoresis) and K_i measurements by NMR-based editing assays.

The promise of transition state analogs for A3 inhibition was foreshadowed by recent biochemical studies and prior studies on *E. coli* CDA. While the reported structure provides a template for the discovery of A3G inhibitors, it is primarily confirmatory. The manuscript is technically solid but dense and difficult to read, possibly better suited for a more specialized biochemistry or structural biology journal.

Some questions the authors may wish to address in a revised manuscript:

1-A3G edits ssDNA containing CC dinucleotides. How will the structural study on A3G with Zeb-DNA shed light on other A3 enzymes such as A3A and A3B that target TC nucleotide motifs?

2-Why is K_d for Zeb-DNA measured by MST (0.27 micromolar) different than the K_i measured by NMR(3 micromolar) reported in Table 1 and 2 respectively?

3-Fig 1, the authors may wish to mention the rotation (and angles) relating the left and right sides of panel B.

4-Fig 4, the depiction of side information from symmetry mates in the *E. coli* CDA structure in panel B and -especially-panel C is confusing; it is unclear if this adds anything to the manuscript so the authors might consider simplifying.

5-In line 45 of the introduction, the authors state "While those A3s play a beneficial role in antiviral immunity, they can also lead to drug resistance." without citing any of the relevant literature. For example, the role of A3G in sublethal mutagenesis of HIV-1 is controversial, so the authors should cite the literature on this point: PMID: 19343218; 25080100; 27186986 ;31025011.

Reviewer #2:

Remarks to the Author:

The manuscript provided by Maiti, et al mainly described the crystal structure of a dZ-ssDNA inhibitor bound to active A3G-CD2 domain. This structure revealed that the activated dZ/H₂O mimics the transition state through coordinating the active site Zn²⁺ and engaging in additional stabilizing

interactions, including the one with the catalytic residues E259. Generally, it is difficult to capture the A3's transition state in the complex structure. Therefore, this structure is very exciting for the researchers in this field. Based on this, I suggested to accept this manuscript after the authors make minor revisions listed below.

(1) In line 57, as to small molecular inhibitors of A3s, within my knowledge, except the references 24, 25 and 26 the authors mentioned, a new small paper, titled as "Aromatic disulfides as potential inhibitors against interaction between deaminase APOBEC3G and HIV infectivity factor", was published at *Acta Biochimica et Biophysica Sinica*, 2022, 54(5), 725-735. The authors should cite it in introduction part of the manuscript.

(2) Remove "}" in line 59.

(3) In lines 70-80, as to the reported structures of A3A and chimeric A3B-CTD with ssDNA, only references 29 and 30? The authors should not exclude structures determined by other techniques, for example, NMR. They should cite other published results as well as ref 29 and 30.

(4) The complex structure in this manuscript is useful for new drug design. However, due to very high A3s' structural similarity, their catalytic deamination mechanisms are almost identical. In the future, how to increase the inhibitor binding selectivity to one member of A3 family? Please add some discussions in the text.

(5) Figure S3, two DNA strands from two complexes' structures are not easily to be distinguished. I suggest to change the blue into other color.

(6) In table 1, can the authors explain the differences of Kd values between CTD2 and CTD2*, and between DNA with dZ and without dZ?

(7) The current structure shows the intermediate state ligates with zinc ion, if in the future, the inhibitor is designed also to ligate with zinc ion, will it produce (enzymatic) toxic effects to all A3 members in vivo?

We would like to thank the reviewers for the time and attention dedicated to helping us improve our manuscript. We answered all comments point-by-point in the following sections. Reviewers' comments are shown in italic font and highlighted in gray. Newly added sentences in the revised manuscript are highlighted yellow.

Reviewer #1 (Remarks to the Author):

Some questions the authors may wish to address in a revised manuscript:

1-A3G edits ssDNA containing CC dinucleotides. How will the structural study on A3G with Zeb-DNA shed light on other A3 enzymes such as A3A and A3B that target TC nucleotide motifs?

We added sentences in Discussion as “Beyond the active site, the rest of the complex between our active and inactive A3G structures display very little conformational change as judged by the low RMSD of the backbone (0.2 Å). Furthermore, all the interactions of specific nucleotides T₋₃, C₋₂, C₋₁ and A₊₁ are essentially conserved in both structures, which suggests that we can accurately predict structural models of A3A or A3B with dZ-ssDNA inhibitors as the A3A:ssDNA and A3B:ssDNA structures are available^{34, 35, 36}. These structural models with the activated dZ/H₂O may be used for designing inhibitors with higher affinity and specificity to A3A or A3B, as the catalytic domain of A3G recognizes nucleotides containing CC, while A3A and A3B recognizes nucleotides containing TC. Since A3A and A3B cause mutations in human genome in cancer cells^{18, 19, 20, 21, 22, 23, 24, 25, 26, 27}, inhibitors, with TdZ, can likely inhibit both A3A and A3B, while avoiding A3G, would be beneficial.”

2-Why is K_d for Zeb-DNA measured by MST (0.27 micromolar) different than the K_i measured by NMR(3 micromolar) reported in Table 1 and 2 respectively?

We added a sentence in the Result section as “While the K_D value (0.27 ± 0.07 μM) and K_i value (3.07 ± 0.78 μM) are not exactly the same, this is likely due to variations in experimental conditions which include varied protein concentration and equilibration time.”

3-Fig 1, the authors may wish to mention the rotation (and angles) relating the left and right sides of panel B.

We added the rotation relating **Fig. 1b** and **Fig. 1c** in the figure caption of **Fig. 1**. We appreciate this reviewer for a good suggestion.

4-Fig 4, the depiction of side information from symmetry mates in the E. coli CDA structure in panel B and -especially-panel C is confusing; it is unclear if this adds anything to the manuscript so the authors might consider simplifying.

We deleted the side chains from symmetry mates from **Fig. 4b** and **Fig. 4c**, and deleted sentences describing the sidechains from symmetry mates in the result section.

5-In line 45 of the introduction, the authors state "While those A3s play a beneficial role in antiviral immunity, they can also lead to drug resistance." without citing any of the relevant literature. For example, the role of A3G in sublethal mutagenesis of HIV-1 is controversial, so the authors should cite the literature on this point: PMID: 19343218; 25080100; 27186986 ;31025011.

We added citations of these literatures, and changed the sentence as "However, while those A3s play a beneficial role in anti-viral immunity, they may have a contribution for leading to drug resistance."

Reviewer #2 (Remarks to the Author):

(1) In line 57, as to small molecular inhibitors of A3s, within my knowledge, except the references 24, 25 and 26 the authors mentioned, a new small paper, titled as "Aromatic disulfides as potential inhibitors against interaction between deaminase APOBEC3G and HIV infectivity factor", was published at Acta Biochimica et Biophysica Sinica, 2022, 54(5), 725-735. The authors should cite it in introduction part of the manuscript.

We added a citation of this literature.

(2) Remove "}" in line 59.

We deleted "}" in line 59.

(3) In lines 70-80, as to the reported structures of A3A and chimeric A3B-CTD with ssDNA, only references 29 and 30? The authors should not exclude structures determined by other techniques, for example, NMR. They should cite other published results as well as ref 29 and 30.

We added a citation of the literature that reported NMR structures of the A3A-ssDNA complex.

(4) The complex structure in this manuscript is useful for new drug design. However, due to very high A3s' structural similarity, their catalytic deamination mechanisms are almost identical. In the future, how to increase the inhibitor binding selectivity to one member of A3 family? Please add some discussions in the text.

Please see the answer for Reviewer #1' question 1. We added sentences in the Discussion in order to response both comments.

(5) Figure S3, two DNA strands from two complexes' structures are not easily to be distinguished. I suggest to change the blue into other color.

We would like to keep blue color for dZ-DNA as it is used in main Figures. Therefore, we changed transparency of ssDNA in the A3G-CTD2* structure. The dZ-DNA and ssDNA may be difficult to distinguish partly because they are well superimposed except the dZ and target cytidine.

(6) In table 1, can the authors explain the differences of Kd values between CTD2 and CTD2, and between DNA with dZ and without dZ?*

CTD2 is catalytically active that catalyzes 2'-deoxy-zebularine (dZ) to generate an intermediate that binds tighter than dZ, therefore K_d of dZ-ssDNA for CTD2 is smaller than that for CTD2* as CTD2* is not catalytically active. Comparison of dZ-ssDNA and real substrate ssDNA (no dZ) affinities for CTD2* (catalytically inactive) indicates that replacement of the target cytidine by dZ made the affinity weaker.

(7) The current structure shows the intermediate state ligates with zinc ion, if in the future, the inhibitor is designed also to ligate with zinc ion, will it produce (enzymatic) toxic effects to all A3 members in vivo?

Small compounds which can chelate Zn may or may not have inhibitory effect for catalytic function of all APOBEC3 protein. As we described in the answer for reviewer #1' comment 1, we are likely to generate A3G or A3A/A3B specific inhibitors using activated dZ/H₂O.